# Enhanced Nitrate Ions Remediation Using Fe^0^ Nanoparticles from Underground Water: Synthesis, Characterizations, and Performance under Optimizing Conditions

**DOI:** 10.3390/ma15145040

**Published:** 2022-07-20

**Authors:** Hany M. Abdel-Lateef, Mai M. Khalaf, Alaa El-Dien Al-Fengary, Mahmoud Elrouby

**Affiliations:** 1Department of Chemistry, College of Science, King Faisal University, P.O. Box 400, Al-Ahsa 31982, Saudi Arabia; mmkali@kfu.edu.sa; 2Chemistry Department, Faculty of Science, Sohag University, Sohag 82425, Egypt; alaa.eldaly2@yahoo.com; 3Faculty of Science, King Salman International University, Ras Sudr 46612, Sinai, Egypt

**Keywords:** zero-valent iron, nanoparticles, nitrate removal, under groundwater treatment, electrochemical sensing

## Abstract

The presence of nitrates in water in large amounts is one of the most dangerous health issues. The greatest risk posed by nitrates is hemoglobin oxidation, which results in Methemoglobin in the human body, resulting in Methemoglobinemia. There are many ways to eliminate nitrates from underground water. One of the most effective and selective methods is using zero-valent iron (ZVI) nanoparticles. ZVI nanoparticles can be easily synthesized by reducing ferric or ferrous ions using sodium borohydride. The prepared ZVI nanoparticles were examined by scanning electron microscopy (SEM), energy dispersive X-ray spectroscopy (EDX), electron microscopy (TEM), X-ray diffraction (XRD), Brunauer–Emmett–Teller (BET) surface area, and zeta potential. We aim to eliminate or reduce the nitrates in water to be at the acceptable range, according to the world health organization (WHO), of 10.0 mg/L. Nitrate concentration in water after and before treatment is measured using the UV scanning method at 220 nm wavelength for the synthetic contaminated water and electrochemical method for the naturally contaminated water. The conditions were optimized for obtaining an efficient removing process. The removal efficiency reaches about 91% at the optimized conditions.

## 1. Introduction

Regarding the qualifications of the World Health Organization (WHO), the presence of nitrate ions in potable water is a significant environmental problem that fatally affects human health. Nitrates are found in groundwater due to animals’ decomposition, nitrogenous fertilizers, industrial processes, and agriculture runoff [1,2,3]. The danger of nitrate to humans is its reduction to nitrite, which is involved in the reduction process of hemoglobin to methemoglobin. So, oxygen cannot be transferred to body tissues, leading to methemoglobinemia [4]. The maximum acceptable value for the concentration of nitrate ions in drinking water is about 45 ppm in the United States, while the World Health Organization (WHO) and the European community have set the maximum convenient concentration to be 50 ppm [5].

The methods of determination of nitrate ions in potable water can be assorted into two types, direct and indirect. In the electrochemical detection method, the system can translate the concentration of nitrate ions to an electrochemical signal such as impedance, current, or potential difference. There are also numerous types of chromatographic techniques, such as ion chromatography (IC), high-performance liquid chromatography (HPLC), and ultra-performance liquid chromatography (UPLC). Moreover, other techniques, such as UV spectrometry, electron capture, fluorimetry, and mass spectroscopy, can be used to determine nitrates [6].

The removal of nitrate (denitrification) from drinking water can be performed via many techniques, such as electrodialysis, reverse osmosis, catalysis, electrocatalytic reduction, a membrane bioreactor, ion exchange, and a biological denitrification process [7]. nZVI is assumed to be one of the most promising and efficient materials used for the denitrification of drinking water. nZVI is a substance with great power and usefulness for ecologically friendly groundwater purification. In practical tests with diverse types of water, nZVI has effectively remediated hazardous organic and inorganic components, including chlorinated hydrocarbons, nitroaromatics, azo dyes, nitroamines, alkali earth metals, some transition metals (such as Cr, Co, Cu, Pb, Ni, Zn, and Cd), metalloids, and also cyanobacteria [8,9,10,11,12]. Moreover, nZVI is a significant reducing material with a reduction potential of −0.45 V (Fe/Fe(II)) [13,14]. Variation of conditions or operational parameters is an important research study. There are many factors affecting the process of nitrate removal using zero-valent iron nanoparticles, such as the pH of a solution, temperature, dissolved oxygen, nitrate concentration [15], and nZVI dosage. The effect of pH has not been studied in this work. This is because pH variation is costly and not a favorite matter in drinking water treatment plants. So, the pH has been used as it is.

nZVI can be synthesized via reduction of ferric [16,17] or ferrous ions [18,19] using borohydride as a reducing agent, but its nanoparticles are very reactive and quickly oxidized to iron oxides. So, it must be saved under the ethanol layer [19,20]. In this work, nZVI has been morphologically distinguished via utilizing TEM, SEM, and zeta potential. Additionally, the phases and the chemical composition have been detected via EDX and XRD, respectively. The contaminated underground water containing nitrate was treated using zero-valent iron nanoparticles that were prepared via a facial method with different dosages at different contact times and different nitrate concentrations to obtain the optimum operating conditions and best results.

## 2. Materials and Methods

### 2.1. Materials

Sodium borohydride and absolute ethanol were supported by Fisher scientific co. Ferric chloride was purchased from El-Nasr co., with potassium nitrate, sodium hydroxide, silver nitrate, potassium chloride, and cobalt nitrate. No further purifications were done for all chemicals used in all experiments.

### 2.2. Synthesis of nZVI

As shown in the equation below, zero-valent iron nanoparticles were produced by reducing ferric ions with sodium borohydride.
6NaBH_4_ + 2FeCl_3_ + 8 H_2_O → 2Fe^0^ + 6 NaCl + 6B(OH)_3_ + 21H_2_(1)
0. 1 M Sodium borohydride was prepared by dissolving about 0.3783 g of NaBH_4_ in 100 mL bi-distilled water. For preparing the ferric solution, 0.5406 g of FeCl_3_·6H_2_O was stirred in a mixture of ethanol and doubly distilled water (4/1 *v*/*v*) till dissolving [21].

For nZVI synthesis, sodium borohydride was transferred to a burette, and FeCl_3_ solution was transferred to a flask; then, additional NaBH_4_ solution was dropped under dynamic stirring until the appearance of a dark black precipitate. After that, the remaining quantity of sodium borohydride was added to the FeCl_3_ solution under stirring for 10 min. A black precipitate of nZVI resulted. The synthesized ZVI nanoparticles were decanted. The remaining ZVI nanoparticles were agglomerated and then filtered, then washed with 25 mL ethanol. To prevent the oxidation of nZVI, it should be saved under ethanol to be used for further experiments and characterization.

### 2.3. nZVI Characterization

The SEM micrograph of ZVI nanoparticles was captured via the JEOL JSM 6510 lv instrument. This instrument detects the texture of crystal growth and the particle size of the studied materials. EDX analysis was achieved using an Oxford X-Max 20 apparatus attached to the scanning electron microscope instrument. Additionally, images of the nZVI particles were recorded with the JOEL Transmission microscope TEM (JOEL Ltd., Tokyo, Japan). XRD analysis of Fe^0^ nanoparticles was performed via the Bruker D8 advanced instrument provided with a copper-sealed tube x-ray source producing Cu kα radiation at a wavelength of about 1.5406 A° from a generator at 40 kV and 40 mA. A suitable quantity of Fe^0^ nanoparticles was placed in a glass holder and captured from 20° to 100°. The specific surface area and the pore size analysis of the nano-sized particle were detected with the BET (the Brunauer–Emmett–Teller isotherm) using St 3 on the NOVA touch 4LX instrument. Nanoparticle samples were pre-dried at ambient temperature (30 ± 1 °C) in an evacuated desiccator, degassed, and then introduced to nitrogen at 77 K at perfectly relative pressure values. Zeta potential (ζ-potential) analysis was performed via a Malvern Zeta-size Nano-zs 90 using a clear disposable zeta cell.

### 2.4. Nitrates Removal by ZVI Nanoparticles

For removing nitrates using ZVI, the reaction was carried out by adding a suitable dosage of ZVI to certain concentrations of nitrates with mechanical stirring using a jar test instrument and with variation of the reaction conditions, such as time and nitrate concentration. The residual nitrate after treatment was determined via spectral and electrochemical methods for synthetic and natural contaminated water, respectively. All determinations were repeated three times to confirm the reproducibility. Normal amounts for potable water for K, Na, Ca, and Mg cations and Cl, SO_4_^2^, HCO_3_, and CO_3_^2−^ anions were present, with a high level of NO_3_^−^ anions. Additionally, the TDS was 300 mg/L. The pH value for the synthetic water was in the normal value of 7.5

## 3. Results and Discussion

### 3.1. ZVI Nanoparticles Characterization

According to the SEM study (Figure 1a,b), the ZVI nanoparticles produced under the optimal conditions are spherical in form and seem gathered together. Most of the prepared ZVI particles are in the nanoscale range, of about an average size of 40 nm. It is also noted that the particles are homogeneously sized and arranged and have the same shade. This may be attributed to the prepared nZVI having a large surface area and it is expected to be a promising material for capturing the nitrate ions in water.

The energy dispersive X-ray analysis (EDX) of the prepared ZVI nanoparticles shows a strong signal in the iron region. This affirms the generation of ZVI iron nanoparticles with a high percentage of about as described in Figure 1c.

The other peaks are due to some impurities in the sample, which may come from the ferric chloride sample. A relatively small signal is attributed to the oxygen formation at about 0.5 keV. This may be attributed to the quantity of iron subjected to air which reacts with the active nZVI after synthesis and transference to the analysis.

TEM analysis provides a good description and definition of the morphology of the prepared nZVI, as well as determining crystal size. The recorded micrographs publicized in Figure 2a,b point out that the existing nanoparticles looked like cluster-shaped blobs in the nanosize range (13–40 nm) with arranged branches coiling each other.

The analysis of the TEM images was performed using ImageJ software, and a considerable histogram (Figure 2c) of the width of nano ZVI particles was obtained, indicating a mean size distribution of 30 ± 2 nm.

The selected area electron diffraction (SAED) pattern displays the crystalline existing iron nanoparticles distinguished by the light circles (Figure 2d). Clear white rings appeared, matching the BCC crystal phase of Fe^0^ with the basic lattice plane of ($\bar{11}$0). The pattern provided four interplanar distance parallels to 2.027, 1.4333, 1.1703, and 1.0135 Å with a lattice constant a = 2.86660 Å without any further crystal structure. This corroborates the successful gaining of zero-valent iron, keeping it out of oxidation.

The XRD pattern of the synthesized zero-valent iron nanoparticles at the optimized conditions is exhibited in Figure 3. The figure demonstrates four main spectral signals that are attributed to zero-valent iron of different phases that appeared at two theta angles of 44.67, 65.018, 82.327, and 98.94°.

The analysis confirms that the considerable crystalline zero-valent iron (Fe^0^) lattice of nZVI as a cubic structure matches the value for the miller indices ($\bar{1}\bar{1}$0), ($\bar{2}0$0), ($\bar{2}\bar{1}\bar{1}$), and ($\bar{2}\bar{2}$0) of bcc Fe^0^ presented in JCPDS (00-003-1050) [22]. The corresponding XRD data of the synthesized ZVI nanoparticles are summarized in Table 1.

The average crystallite size of the synthesized nZVI was determined using XRD via the Scherrer equation [23] as follows:(2)$$D=\frac{0.9\;ƛ}{\beta \cos\theta }$$
where *D* is the size of particle in nm, *θ* is the Bragg angle obtained from 2*θ* and corresponds to the peak intensity at the maximum, and *β* is the full width at half maximum (FWHM), [24,25]. The mean value of the calculated crystal size is found to be 10.23 nm.

The estimated surface area S_BET_ of nZVI was found to be 133.987 m²/g based on the classic BET method with a correlation coefficient of 0.998868. Similar approaches distinguished the surface texture of zero-valent iron as a type II adsorption isotherm that was well matched with the prepared nZVI sample [22,26]. The total pore volume was calculated to be 0.033 cm^3^ g^−1^ for the prepared nZVI nanoparticles. This high value of S_BET_ and moderate pore volume values declared that the followed synthesis route meritoriously suppressed the accumulation of nZVI particles. Additionally, the obtained S_BET_ value of nZVI constructed additional active sites, and the great pore volume was helpful to the quick dispersion of contaminants that augment the removal of nitrate ions.

Zeta potential is an effective tool for assessing the intensity of electrostatic attraction or repulsion between particles in a suspension. Furthermore, it is a key variable in examining the stability of nano-sized particles in aqueous environments. Nanoparticles with zeta potentials larger than +30 mV and lower than -30 mV are considered stable for liquid suspension without steric stabilization. However, in non-colloidal suspensions, this value can be altered [27]. The ZVI nanoparticles’ stability was assessed here with a zeta potentiometer after synthesizing the metal nanoparticles (Figure 4). The observed zeta potential for the suspended nZVI is found to be 0.288 mV. This value indicates that the ZVI is in the range of nano-sized particles, as confirmed by XRD, SEM, and TEM analysis and its stability.

### 3.2. Effect of Reaction Conditions

#### 3.2.1. Effect of ZVI Nanoparticles Dosage

For studying the effect of nZVI dosages, an initial nitrate concentration of about 100 ppm was used, and the contact time was 30 min at ambient temperature (25 °C). The dosage quantity of nZVI varied from 0.5 to 3.0 g/L. It’s clear from Figure 5A that, by increasing the nZVI dosages, the remaining nitrate concentration after treatment is decreased to about 94.77% of the removal efficiency at 2.5 g/L of nZVI dosage, which corresponds to 5 ppm (from the initial concentration of nitrate of 100 ppm). After that, the removal efficiency slightly decreased to 93.54% at 3 g/L nZVI dosages. These values are below the range of the WHO; the greatest dosage of nitrate is 50 ppm [5]. This reflects the high efficiency of the prepared nZVI at our simple optimized conditions. The chemical reaction between nZVI and nitrate ions can be described as follows;
4Fe^0^ + NO_3_^−^ + 7 H_2_O → 4Fe^2+^ + NH_4_^+^ + 10 OH^−^(3)

The remaining concentration of nitrates after the treatment process versus the dosage of nZVI at the conditions mentioned above is illustrated in Figure 5B. It can be noted from the figure that the remaining concentration of nitrates decreases sharply from about 38 to 12 ppm at the doses of 0.5 and 1.0 g/L, respectively. After that, it decreases slightly from 12 to 5 ppm at the dose of 2.5 g/L nZVI. Then it slightly increases to about 7 ppm for the dose of 3 g/L ZVI.

#### 3.2.2. Effect of Initial NO_3_ Concentration

The effect of nitrate concentration on the removal efficiency of nZVI has been investigated to detect the maximum removal capacity for the studied material, as shown in Figure 6. Here, 1.0 g/L of nZVI dosage was used, and nitrate’s initial concentration varied from 50 to 110 ppm. It can be noted from Figure 6A that the efficiency of nitrate removal ranges between 90.0 and 94.0% for all concentrations except that of 110 ppm, at which it reaches 83.0%. This can be attributed to the covered active sites decreasing the capacity of the nZVI. However, the obtained values of the remaining nitrates after treatment are still below the maximum value of nitrate concentrations of the WHO, even if using small dosages.

From Figure 6B, it is noticed that for 1.0 g/L ZVI dosage, there was a saturation of active sites, so that at a nitrates concentration of 110 ppm, the removal efficiency was decreased, as confirmed above.

#### 3.2.3. Effect of Reaction Contact Time

The influence of contact time on the remediation efficiency of nZVI has been studied to detect its efficiency in the removal process. Here, 1.0 g/L nZVI dosage and 100 ppm nitrate concentration was used with varying of the contact time. ranging between 10 and 50 min, as shown in Figure 7A. It can be noted that all contact times are suitable for use, as all remaining nitrate concentrations are under the maximum acceptable value.

From Figure 7B, it is observed that, with increasing of the contact time, the nitrate removal efficiency is increased until reaching 91.4% at minute 30. Then, there was no noticeable change after minute 50. This means that all active sites were occupied and saturated at 30 min and confirms the fast removal process favorable for industrial and commercial purposes.

### 3.3. Electrochemical Synthesis of AgPtCo onto GCE for Sensing of Nitrate in Natural Water

It should be mentioned that the need for an efficient and selective sensor for the residual nitrate ions becomes a necessary demand, especially in the naturally contaminated sample, due to interference. Cyclic voltammetry of Ag (as a major element), Pt, and Co ions (as minor elements) was performed to optimize the suitable conditions for the electrodeposition process as a new nitrate sensor. Figure 8a exhibits the cyclic voltammetric attitude of the electroreduction of Ag, Pt, and Co ions on GCE at a ramp rate of 50 mV/s, at ambient temperature and slightly acidic media (pH = 5). The bath of electroreduction consists of 0.01 M AgNO_3_, 0.002 M CoCl_2_.6H_2_O, 0.002 platinic acids, and 0.1 M KCl as a supporting electrolyte. The figure exhibits that the electrochemical potential reduction peaks for the Ag, Pt, and Co ions are at +0.29, −0.20, and −0.50 V for Ag, Pt, and Co ions [28,29], respectively.

The potentiostatic method was utilized for achieving the electrochemical deposition of the proposed nitrate sensor (Ag, Pt, Co). Furthermore, the crystal growth’s features and mechanism during the electrochemical synthesis process were examined. The current–time curve was obtained at the optimized potential of deposition at −0.6 V (estimated from the CV behavior) for 600 s at ambient temperature, as shown in Figure 8b.

According to Figure 8b, the current climbs swiftly to its greatest value and then progressively increases over time. The quick increase in the cathodic side current at the start of the current could be interpreted as an increase in the nucleation sites or a construction of a novel phase. The nuclei form diffusion zones during the deposit development stage. These zones obstruct hemispherical mass transference and allow linear mass transfer to occur, leading to a more efficient, smooth surface.

The crystal formation and nucleation mechanism may be identified by comparing the experimental results to the Scharifker-Hills equations [30]. According to the proposed model, nucleation mechanisms may be classified into two types: instantaneous and progressive. The following equations represent the two assumed mechanisms for the instantaneous and progressive, respectively:(4)$${ [\frac{I\left(t\right)}{I_{max}}]}^{2}=\frac{1.9542}{\frac{t}{t_{max}}}{\left\{1-\exp [-1.2564\frac{t}{t_{max}}]\right\}}^{2}$$
(5)$${ [\frac{I\left(t\right)}{I_{max}}]}^{2}=\frac{1.2254}{\frac{t}{t_{max}}}{\left\{1-\exp [-2.3367{\frac{t}{t_{max}}}^{2}]\right\}}^{2}$$
where *t*_max_ and *I*_max_ are the maximum value of time in second (s) and the maximum current in Amper (A).

Figure 8c exhibits the non-dimensional I^2^I^−2^_max_ vs. t.t^−1^_max_ plots obtained from the potentiostatic electrodeposition of Ag, Pt, and Co metals data. The lines colored black and red reflect theoretical data for progressive and instantaneous electrochemical deposition, whereas the blue colored line represents the obtained results. In the first stage, the experimental data for the electrochemical deposition of the given sensor is convenient with the nucleation of the instantaneous model and deviated after running the time.

Figure 9 shows cyclic voltammograms of the modified GCE with AgPtCo immersed in a solution consisting of 0.1 M NaOH as a supporting electrolyte scanned within the range of −0.15 to −1.20 V at a scan rate of 50 mV/s at different concentrations of nitrate ions as labeled in the figure.

The electroreduction of nitrate ions on bare GCE was not noticed, while the electrochemical reduction of nitrate ions is well defined on our modified sensor at a potential of about −1.0 V. This reveals that the electrochemical reduction process of nitrate ions is facilitated via the fabricated efficient electro-catalyst system of AgPtCo/GCE. It is also noted that (Figure 9. inset) there is a strong relationship between concentration and the electroreduction peak current, with a linear regression of 0.99. From this relation, the nitrate ions concentration in natural water in the two wells was detected before and after treatment. The natural nitrate ions concentration was 37.5 ppm and 90 ppm in the first and second wells, respectively. After treatment of the two samples at the optimized condition (as detected before), the nitrate ions concentration was found to be 4 and 5 ppm, respectively. This confirms the suitability of our nitrate removal system under the optimized conditions.

### 3.4. Comparative Studies with the Previous Approaches

Nitrate removal from water can be carried out by various methods but achieving the optimum conditions for the removal process is really the challenge. There are many techniques and strategies for nitrate reduction, including electrochemical methods, biological denitrification, ion exchange, chemical removal, and other methods. Table 2 displays the comparison between some of these methods for the removal process using different materials, indicating nitrate removal efficiency under some conditions [31,32,33,34,35,36,37,38,39]. From the table, it is clear that nitrate removal using ZVI nanoparticles is the best, with a removal efficiency of about 94.77% at a contact time of about 30 min with an initial concentration of nitrate equal to 100 mg per liter.

## 4. Conclusions

It can be concluded that effective zero-valent iron (ZVI) nanoparticles were successfully synthesized via a facile and fast method by reducing ferric ions using sodium borohydride. The prepared ZVI was characterized by SEM, EDX, TEM, XRD, and zeta potential, revealing its stability, nano-sized purity, and activity. The reaction conditions were optimized for obtaining an efficient removing process. The use of 1.0 g/L of nZVI at a contact time of 30 min at ambient temperature for naturally and artificially contaminated water at neutral pH reduces the concentration of nitrate ions ranging from 50 to 110 ppm to be less than the concentration of nitrates in potable water to be within the acceptable range of the World Health Organization (WHO). Additionally, in this work, we developed a new sensor (AgPtCo/GCE) that accurately measured the nitrate ions in natural water before and after treatment.

## Figures and Tables

**Figure 1 materials-15-05040-f001:**
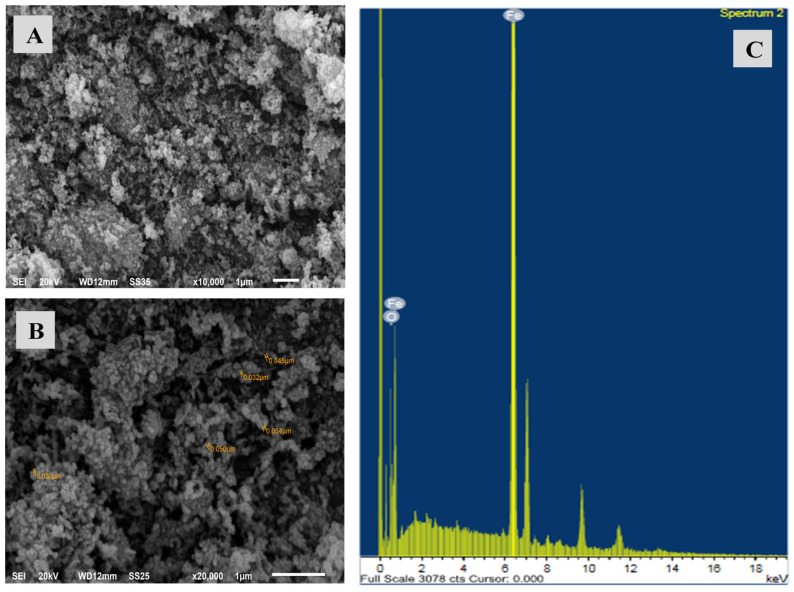
SEM micrographs at two different magnifications (**A**,**B**); EDX analysis of the prepared ZVI nanoparticles at optimum conditions (**C**).

**Figure 2 materials-15-05040-f002:**
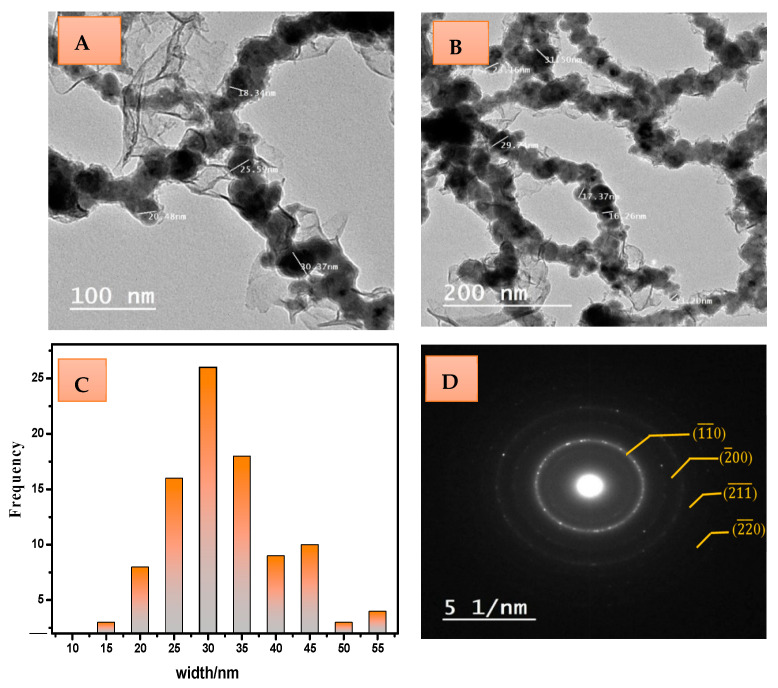
TEM images at different magnifications (**A**,**B**); histogram of the nanoparticle width (**C**); and SAED image (**D**)of nZVI sample.

**Figure 3 materials-15-05040-f003:**
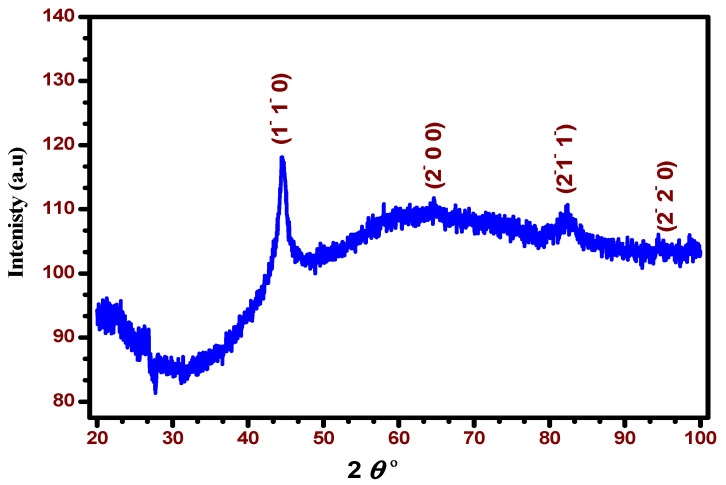
XRD pattern of ZVI nanoparticles synthesized at optimum conditions.

**Figure 4 materials-15-05040-f004:**
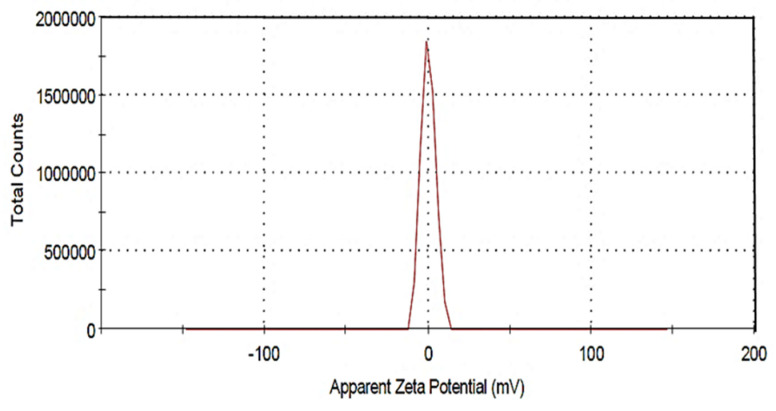
Zeta -potential distribution for ZVI nanoparticles suspension.

**Figure 5 materials-15-05040-f005:**
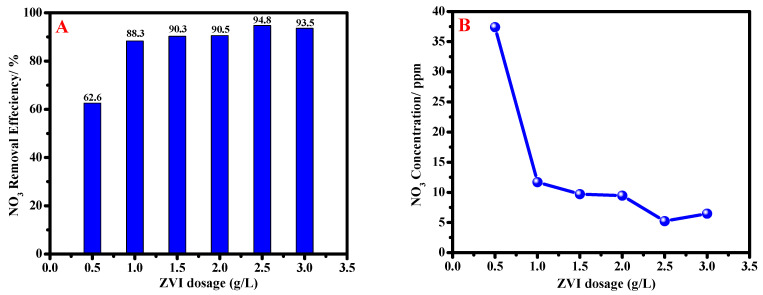
Nitrates removal efficiency versus the dosage of nZVI (**A**) and Nitrate concentration after treatment with different ZVI dosages (**B**) at ambient temperature and 30 min contact time.

**Figure 6 materials-15-05040-f006:**
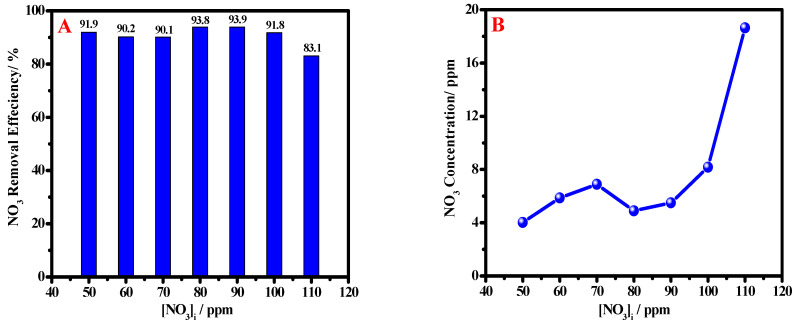
Removal efficiency with varying of the NO_3_ initial concentration (**A**) and initial nitrate concentrations against the remaining concentration in ppm after treatment (**B**) at ambient temperature at 30 min contact time.

**Figure 7 materials-15-05040-f007:**
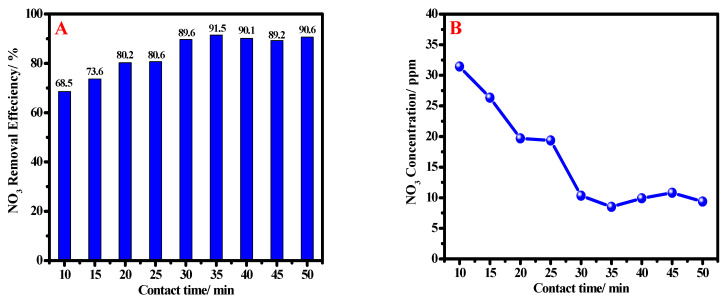
The removal efficiency of nZVI (1.0 g/L) with various contact times (**A**) and dependence of nitrate concentration after treatment on contact time (**B**) at ambient temperature.

**Figure 8 materials-15-05040-f008:**
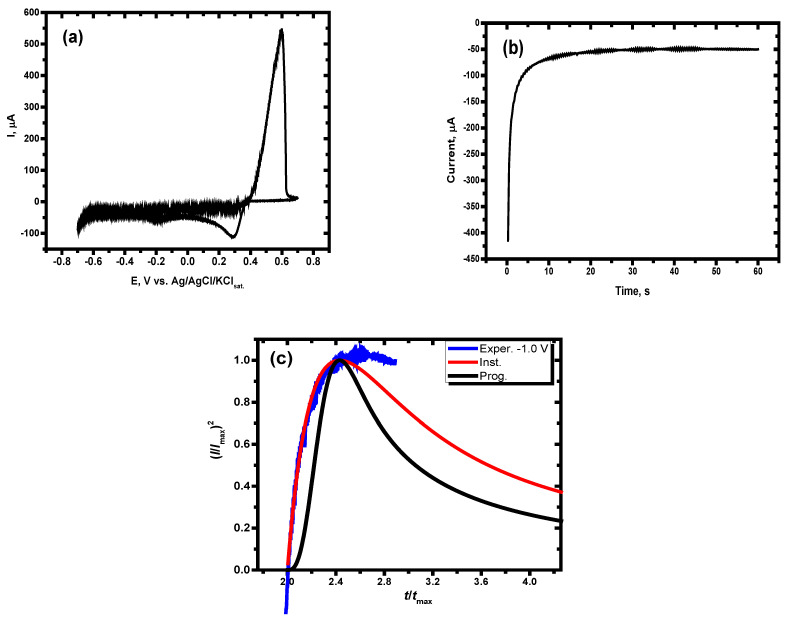
(**a**) Cyclic voltammetry of Ag, Pt, and Co ions as shown, (**b**) chronoamperometric behavior at E_d_ = −0.60 V, and (**c**) (*I*/*I*_max_)^2^ vs. (*t*/*t*_max_) relationship for experimental and theoretical data of the instantaneous and progressive crystal growth.

**Figure 9 materials-15-05040-f009:**
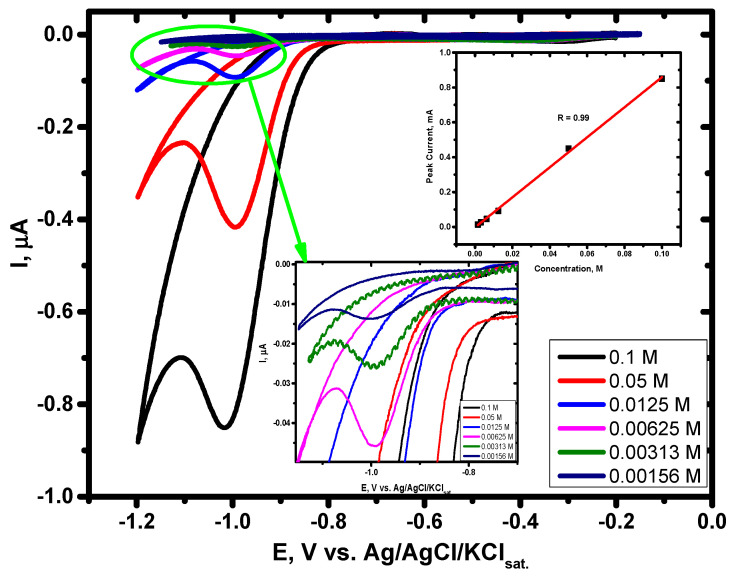
CV of 0.1 M NaOH on AgPtCo/GCE at 50 mV/s, at different nitrate ions concentrations. The insert figure is the relation between nitrate concentration in M and the electrochemical reduction peak current.

**Table 1 materials-15-05040-t001:** XRD data of the synthesized nZVI at the optimum conditions.

d	2*θ*	I fix	h	k	l
2.02700	44.670	999	−1	−1	0
1.43330	65.018	130	−2	0	0
1.17030	82.327	220	−2	−1	−1
1.01350	98.935	67	−2	−2	0

**Table 2 materials-15-05040-t002:** Optimum parameters for nitrate removal by different adsorbents and methods.

No.	Method	NO_3_ Initial Conc.	NO_3_ Removal Efficiency	Time	Refs.
1	zero-valent iron nanoparticles	100 mg/L	94.77%	30 min	This work
2	sulfur/pyrite-based bioreactor	50 mg/L	99.20%	---	[31]
3	polyoxometalates (POM)/TiO_2_/Cu	30 mg/L	76.53%	6 h	[32]
4	Ti/PbO_2_ anode	50 mg/L	47.70%	120 min	[33]
5	Chlorella vulgaris	2322 mg/L	85.60%	17.2 h daily photoperiod in a 13-day culture	[34]
6	electrocoagulation	-----	88.48–94.1%	120 min	[35]
7	semi and fully continuous-flow Donnan dialysis systems	-----	80%	4 h	[36]
8	capacitive deionization with activated carbon/PVDF/polyaniline/ZrO_2_ composite electrode	70 mg/L	60.01%	62 min	[37]
9	hypothermia bacterium Pseudomonas putida Y-9	less than 100 mg/L	82.00%	------	[38]
10	aerobic denitrifying actinomycete Streptomyces sp. XD-11-6-2	-----	72.29%	------	[39]

## Data Availability

The raw/processed data generated in this work are available upon request from the corresponding author.

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
