# Peer review of "Enhanced Nitrate Ions Remediation Using Fe0 Nanoparticles from Underground Water: Synthesis, Characterizations, and Performance under Optimizing Conditions"

_materials, 2022, doi:10.3390/ma15145040_

Round 1

Reviewer 1 Report

The article highlights the current topic. The list of literature has a very wide range of studies. The graphics are quite intersting. Research error is not described. It was possible to apply the apparatus of neural networks for modeling. The authors poorly described modern modeling methods and modeling apparatus. The mathematical description is poorly done.

Author Response

Reviewer 1

The article highlights the current topic. The list of literature has a very wide range of studies. The graphics are quite intersting. Research error is not described. It was possible to apply the apparatus of neural networks for modeling. The authors poorly described modern modeling methods and modeling apparatus. The mathematical description is poorly done.

Author response:

Thank you for your useful comments. The results were repeated three times and are reproducible. The modeling in this work is focused on the electrodeposition of the electrode fabrication to know the nucleation mechanism, and this method is highly recommended and reported in those cases. As described, the crystal formation and nucleation mechanism were identified by comparing the experimental results to the Scharifker-Hills equations showing that in the proposed model, nucleation mechanisms may be classified into two types: instantaneous and progressive.

Reviewer 2 Report

This manuscript falls within the scope of the journal. However, I think that the manuscript will need revision before the work can be published.  

1. Introduction the structure needs to be adjusted, and it is suggested to be divided into three paragraphs for description (nitrate pollution, harm, remediation technology). 

2. There are many format and symbol problems in the paper. 

3. The format of the form needs to be unified. 

4. Figures are too rough and needs to be remade. 

5. The conclusion is too simple and needs to be refined. 

6. The manuscript needs to be polished by professionals.

Author Response

Reviewer 2

This manuscript falls within the scope of the journal. However, I think that the manuscript will need revision before the work can be published.

Author response:

Thank you for your valuable comments.

  1. Introduction the structure needs to be adjusted, and it is suggested to be divided into three paragraphs for description (nitrate pollution, harm, remediation technology).

Author response:

Thank you for your valuable comments. The structure of the introduction is now reformulated and divided into four paragraphs. The first paragraph deals with the nitrate harmful and its suitable concentration, and the second one illustrates the determination of nitrate-based on different methods. The third paragraph elucidates the various techniques of nitrate removal, including zero-valent iron, and the last paragraph clarifies the synthesis and characterization of zero-valent iron.

  1. There are many format and symbol problems in the paper.

Author response:

Thank you for your valuable comments. All formats and symbols have been corrected in the revised paper.

  1. The format of the form needs to be unified.

Author response:

Thank you for your valuable comments. All done in the new version of the revised manuscript.

  1. Figures are too rough and needs to be remade.

Author response:

Thank you for your valuable comments. All figures are adjusted.

  1. The conclusion is too simple and needs to be refined.

Author response:

Thank you for your valuable comments. The conclusion is now extended and modified in the new version of the manuscript.

  1. The manuscript needs to be polished by professionals.

Author response:

Thank you for your valuable comments. The manuscript has been polished as in the new version.

Reviewer 3 Report

Enhanced nitrate ions remediation using Fe0 nanoparticles from underground water: Synthesis, characterizations, and performance under optimizing conditions is very interesting paper. Some improvement is required.

Line 18: We aim to eliminate or reduce the nitrates in water to be at the acceptable range, according to the world health organization (WHO). What is the value (mg/L) according to WHO.

Line 35, 36: The maximum acceptable value for the concentrations of nitrate ions in the drinking  water is about 45 ppm in the United States, while the World Health Organization (WHO) and the European community have set the maximum convenient concentration to be 50 ppm. In which volume of solution: 45 ppm/L (?); 50 ppm/L

The U.S. Environmental Protection Agency (EPA) standard for nitrate in drinking water is 10 milligrams of nitrate (measured as nitrogen) per liter of drinking water (mg/L).  Please to check this value.

Line 51_ Zero valent iron nanoparticles (nZVI) are assumed to be one of the most promising and efficient materials used for the denitrification of drinking water. Please to add morphological characteristics mostly used Zero valent iron nanoparticles (particle size, form, surface area).

Line 94: The nanoparticles of ZVI were filtered. Which type of filter was used in order to collect very fine nanoparticles?

Line 227, 228:  It can be noted from the figure that the remaining concentration of nitrates decreases sharply from about 38 to 12 ppm at the doses of 0.5 and 1 g/l, respectively. What is calculated stoichiometrically required concentration of nanosized iron for this removal of nitrate ions,

General question:

1. what is chemical reaction for the removal of nitrate ions using iron.

2. What is chemical composition of solution (underground water) for nitrate removal with nanosized iron

3. What was pH-VAlue of water solution

Author Response

Reviewer 3

Enhanced nitrate ions remediation using Fe0 nanoparticles from underground water: Synthesis, characterizations, and performance under optimizing conditions is very interesting paper. Some improvement is required.

Author response:

Thank you for your valuable comments.

  1. Line 18: We aim to eliminate or reduce the nitrates in water to be at the acceptable range, according to the world health organization (WHO). What is the value (mg/L) according to WHO.

Author response:

Thank you for your valuable comments. According to the world health organization (WHO), the acceptable value is 10 mg/L. this value has been included in the new version of the manuscript.

Line 35, 36: The maximum acceptable value for the concentrations of nitrate ions in the drinking  water is about 45 ppm in the United States, while the World Health Organization (WHO) and the European community have set the maximum convenient concentration to be 50 ppm. In which volume of solution: 45 ppm/L (?); 50 ppm/L

Author response:

Thank you for your comment, ppm unit is equivelant to mg/L, which means the total volume of solution is 1 Litre.

The U.S. Environmental Protection Agency (EPA) standard for nitrate in drinking water is 10 milligrams of nitrate (measured as nitrogen) per liter of drinking water (mg/L). Please check this value.

Author response:

The value has been well checked. The acceptable value is 10 mg/L, and the maximum value is 50 mg/L.

Line 51_ Zero valent iron nanoparticles (nZVI) are assumed to be one of the most promising and efficient materials used for the denitrification of drinking water. Please add morphological characteristics mostly used Zero valent iron nanoparticles (particle size, form, surface area).

Author response:

The destination of this sentence is that ZVI nanoparticles compared with other materials are considered the most efficient. Also, ZVI in the metallic powder form is considered a reducing agent and reacts with nitrate to produce the reduced form of nitrate. The reduced form depends on the efficiency of the reducing agent. The efficiency of ZVI depends on the size and surface area. These parameters are included and discussed in the characterization part. 

Line 94: The nanoparticles of ZVI were filtered. Which type of filter was used in order to collect very fine nanoparticles?

Author response:

Thank you for your notice, the synthesized ZVI nanoparticles were first decanted. The remained ZVI nanoparticles were agglomerated and then filtered.

Line 227, 228:  It can be noted from the figure that the remaining concentration of nitrates decreases sharply from about 38 to 12 ppm at the doses of 0.5 and 1 g/l, respectively. What is calculated stoichiometrically required concentration of nanosized iron for this removal of nitrate ions,

Author response:

The stoichiometrically required amount of zero-valent iron can be easily calculated via the following reaction;

4FeËš + NO3- + 7 H2O → 4Fe+2 + NH4+ + 10 OH-

Four moles of iron are required to remove one mole of nitrate. But in our research, we experimentally use the effect of iron dose on the efficiency and rate of nitrate removal.

  1. what is chemical reaction for the removal of nitrate ions using iron.

Author response:

The chemical reaction was included in the new version of the manuscript, as follows;

 4FeËš + NO3- + 7 H2O → 4Fe+2 + NH4+ + 10 OH-

  1. What is chemical composition of solution (underground water) for nitrate removal with nanosized iron

Author response:

The chemical composition of the natural water is as follows;

Normal amounts for potable water for K, Na, Ca, Mg cations and Cl-, SO42-, HCO3-, CO32- anions, with a high level of NO3- anions. As well as, the TDS is 300 mg/L. These values were included in the revised manuscript.

  1. What was pH-VAlue of water solution

Author response:

The pH value for synthetic water was in the normal value of 7.5 and it was included in the refined manuscript.

Round 2

Reviewer 2 Report

The manuscript has made great progress after modification, but the format and quality of the image need to be optimized.

Author Response

We would like to thank the reviewer for his great efforts and giving useful criticism to the article. Below are answers to each point.

The manuscript has made great progress after modification, but the format and quality of the image need to be optimized.

Author response:

Thank you for your valuable comments. The format and quality of the figures was modified in the revised manuscript and the high qulity images was used.
